# The Effect of Kinesio Taping on Balance and Dynamic Stability in College-Age Recreational Runners with Ankle Instability

**DOI:** 10.3390/healthcare11121749

**Published:** 2023-06-14

**Authors:** Hisham M. Hussein, Walid M. Kamel, Ehab M. Kamel, Mohamed R. Attyia, Tolgahan Acar, Raheela Kanwal, Ahmed A. Ibrahim

**Affiliations:** 1Department of Physical Therapy, College of Applied Medical Sciences, University of Hail, Ha’íl 81451, Saudi Arabiar.sheikh@uoh.edu.sa (R.K.); a.abdalmoniem@uoh.edu.sa (A.A.I.); 2Department of Basic Sciences for Physical Therapy, Faculty of Physical Therapy, Cairo University, Giza 12613, Egypt; 3Critical Care Department, Faculty of Medicine, Cairo University, Giza 11435, Egypt; 4Department of Public Health, College of Public Health and Health Informatics, University of Ha’íl, Ha’íl 81451, Saudi Arabia; em.ahmed@uoh.edu.sa

**Keywords:** recreational runners, kinesio taping, balance, ankle, instability

## Abstract

(1) Background: Running is one of many sports that have increased in popularity since it can be conducted at any time or anywhere. Ankle instability is a common injury that usually occurs during running and is usually associated with abnormalities in postural stability. Recently, kinesio taping has gained increasing interest as a tool that can be used in rehabilitation, to improve stability, and to help in injury prevention. This study aimed to investigate the effect of Kinesio taping on balance and dynamic stability in recreational runners with ankle instability. (2) Methods: This randomized controlled trial recruited 90 RRs with ankle instability. The participants were randomly divided into three equal groups: a KT group (KTG) who received Kinesio taping on their ankle joints; a mixed group (MG) who received Kinesio taping and exercises; and an exercise group (EG) who received exercises only. Outcome measures (balance and dynamic stability) were assessed before and after the end of an 8-week treatment program using a Biodex balance system and a star excursion balance test, respectively. (3) Results: Within-group comparisons showed statistically significant improvements in most of the outcome values when compared to baseline. Overall stability index was statistically significantly better (with a high effect size) in the MG compared to KTG or EG (*p* = 0.01, Cohen’s *d* = 1.6, and *p* < 0.001, Cohen’s *d* = 1.63, respectively). A similar finding was evident in the anteroposterior stability index (*p* = 0.02, Cohen’s *d* = 0.95, and *p* < 0.001, Cohen’s *d* = 1.22, respectively). The mediolateral stability index of the KTG was statistically significantly better with a high effect size when compared to MG or EG (*p* = 0.04, Cohen’s *d* = 0.6, and *p* < 0.01, Cohen’s *d* = 0.96, respectively). The star excursion balance test values were statistically significant with high effect sizes in the posterior (*p* = 0.002, Cohen’s *d* = 1.2) and lateral (*p* < 0.02, Cohen’s *d* = 0.92) directions in the MG compared to KTG and EG. (4) Conclusions: Kinesiotape with exercises is superior to either kinesiotape alone or exercises alone in improving postural stability indices and dynamic stability in recreational runners with ankle instability. Recreational runners with ankle instability should be educated about practicing balance exercises and applying kinesiotape.

## 1. Introduction

Running is a popular recreational sport. It has been practiced by both sexes in different age groups [1]. In European countries, between 12.5 and 31% of the population practice running for pleasure with no fixed distance and probably do not participate in any running competitions; hence, they are referred to as recreational runners (RRs) [2]. This sport has gained popularity because of multiple advantages: it does not need a special place or equipment for practice, and it has great beneficial effects on physical and mental health, which might decrease the mortality rate by 25–40% [3]. Moreover, running has been considered a contributing factor in normalizing lean/fat body mass, improving resting heart rate, enhancing cardiovascular fitness, and helping in the cessation of smoking [4].

Subjects practicing recreational running might be at high risk for running-related injuries. As reported by Ellapen and colleagues, 6.3 injuries occur per 1000 training hours. This incidence rate is higher than that reported in other sports [5]. According to other reports, a 25–85% incidence rate of injuries has been associated with running activities [6]. 

The ankle joint occupies second place, after the knee joint, as the most vulnerable joint to injury during running [5]. While most running-related injuries have a gradual onset due to overuse, sudden ankle injuries can also occur [7]. One of the most common ankle injuries is a lateral sprain [8]. This injury may lead to trauma to the soft tissues and bony components, which consequently affect the function of the injured joint and predispose to injury recurrence and ankle instability. 

Ankle instability has been described as a sustained symptom after a significant ankle sprain [9]. Due to the high prevalence of lateral ankle sprains (LAS), they have been considered the main predisposing factor for ankle instability. LAS has been linked to a decrease in postural stability and neuromuscular control, which affects somatosensory functioning and increases the incidence of recurrent ankle injuries [10]. Postural stability is one of the major functions that depends mainly on cues originating from proprioceptors [11]. Cutaneous stimulation, especially skin stretching, has been considered an influential factor in joint proprioception. More accurate joint position sense was reported when the skin was stimulated [12]. Consequently, any therapeutic intervention that can stimulate skin receptors might have positive effects on joint proprioception, enhance sensorimotor functioning, and therefore decrease the risk of ankle sprains and consequent ankle instability.

Lack of appropriate therapeutic intervention might lead to a recurrence of injury and decreased ankle stability. Subjects who experience ankle instability usually develop a fear of participating in sports due to the feeling of weakness in the ankle joint and their higher susceptibility to recurrent injuries [13].

Kinesio taping (KT) is the application of an adhesive and elastic cotton tape to the skin. This tape is anti-allergenic and can be left in place for three to five days without adverse effects. This therapeutic method has gained increasing interest in the fields of rehabilitation [14], sports [15], and even among healthy subjects [16]. Under taping, the skin is stretched, stimulating cutaneous aspects of proprioception, which, in turn, might affect somatosensory function, muscle strength, and postural stability. Such improvements in proprioceptive function enhance ankle joint performance and decrease the risk of ankle instability [15]. 

The literature contains contradictory evidence regarding the effectiveness of KT. While a previously published systematic review reported limited scientific evidence for KT in the rehabilitation of musculoskeletal injuries [17], others observed favorable effects. Griebert and colleagues applied KT to a group of subjects with medial tibial stress syndrome, a syndrome that results from repeated microtrauma to the tissues and is usually reported in physically active persons due to recreation running; this study reported a significant decrease in the loading stress after applying KT [18]. Significant improvements in strength, postural control, and dynamic and static balance were observed following KT in participants with ankle instability [19]. On the other hand, Hadadi et al. reported no improvement in balance after taping unstable ankles [20]. 

Most of the studies were conducted on healthy subjects or athletes who usually undertake regular fitness training; however, there is a lack of investigations in non-athletic populations who are practicing recreational running.

Therefore, the purpose of this study was to assess the effect of KT on balance and dynamic stability in RRs with ankle instability.

## 2. Materials and Methods

### 2.1. Trial Design

This study was a double-blinded (participants and assessor) randomized controlled trial. The measured outcomes were postural stability indices (measured by a Biodex Balance System) and dynamic stability measured by star excursion balance tests (SEBTs). The outcomes were assessed at two time points: at baseline and after 8 weeks of intervention.

### 2.2. Participants

The authors used social media and written announcements to recruit 90 adults (age ≥ 17 years) from the community of Ha’il City after fulfilling the eligibility criteria. The study was conducted at the University of Ha’il, Kingdom of Saudi Arabia, from 30 November 2022 to 14 March 2023. This study was conducted according to the guidelines of the Declaration of Helsinki and approved by the Institutional Ethics Committee of the University of Ha’il (No.: RG-20222 and date: 9 September 2020). Additionally, the trial was registered on https://clinicaltrials.gov (No.: NCT05709808). The participants signed a written consent form before the start of the study.

The participants were randomly assigned into three groups (n = 30 for each group): the KT group (KTG) received ankle KT only; the mixed group (MG) received KT plus a standard ankle exercise program; and the exercise group (EG) received a standard exercise program only.

The inclusion criteria were: (1) healthy college-age males; (2) BMI in the normal or overweight category; (3) regular participation in running activities (1–3 times/week for the last 6 months) [21]; (4) past history of at least one LAS; (5) ankle instability with a score within 24–27 according to the Cumberland Ankle Instability Tool (CAIT) [22]. Exclusion criteria were: (1) participating regularly in any sport other than recreational running; (2) having vestibular system problems; (3) having a history of traumatic brain injury in the last three months; (4) having a fracture or surgery to a lower limb; and (5) taking drugs that affect the balance or vestibular system.

### 2.3. Interventions

#### 2.3.1. Kinesio Taping

KT was applied to both the KTG and MG by a well-qualified physical therapist. We used the tapping technique performed previously by Mohammed and colleagues [23]. This taping technique was designed to guard against LAS and prevent a recurrence of injury by providing support. In the current study, we applied the KT to the unstable ankle. The participant’s foot was placed in a relaxed and elevated position. The first strip of KT (120% stretch) was applied from the anterior midfoot to a point just inferior to the tuberosity of the tibial bone so that it covered the tibialis anterior muscle. The second strip extended from the medial malleolus, across the heel, and split into 2 branches distal to the lateral malleolus, where the first and second branches covered the anterior and posterior parts, respectively, and reached to the lateral malleolus; from there onwards, both were placed to attach laterally to the end of the first strip of KT. Using a 140% stretching force, a third KT strip was placed across the ankle to cover both malleoli. The fourth strip extended from the arch and stretched to a point located six inches above both malleoli. The Kinesio tape for all participants in the KTG and MG was applied by the same researcher (Figure 1). Once applied, the KT was removed every four days; the area was examined for possible irritation, cleaned with alcohol, and then the KT was applied again for another 4 days. This procedure was repeated until the end of the study period. The participants were instructed to remove the tape once any signs of irritation (itching and/or skin rash) were observed.

#### 2.3.2. The Standard Exercise Program

This type of intervention was performed by MG and EG, according to [24]. The ankle exercises were conducted for eight weeks, three times/week, with each session lasting 60 min. The session started with 10 min of treadmill walking on an even surface at 0.8 m/h, then general stretching exercises for hip, knee, and ankle muscles, proprioceptive training, and finally, each participant finished the session with five minutes of walking slowly. 

The proprioceptive exercises were performed while the participant stood on a Wobble board for static and dynamic balance training with the eyes open, while the last stage would be repeated with the eyes closed. The exercises were divided into three sets: the first set consisted of stages 1–4; the second set, stage 5, was performed with eyes open; and finally, in the third set, stage 5 was repeated with eyes closed. Each set was repeated 10 times, with a 10-s rest between each set. 

First set for proprioceptive exercises:The board was rocked forward and backward.The board was rocked from side to side.Then, with the feet wide apart, the board was rocked in a circulating movement.Stages 1–3 were repeated, but with the knees slightly bent and the hands on the buttocks.

Second set:5.The participants stood on their injured legs and kept the board level for 10 s.

Third set:6.If, in stage 5, the participant could maintain their balance without losing the stability of the board, then stage 5 was repeated with the eyes closed.

### 2.4. Outcomes

#### 2.4.1. Postural Stability Assessment

Demographic data (age, sex, weight, height, BMI, and lower limb dominance) was reported during the first meeting. Limb dominance was determined by asking the participants the following question: “If you have a ball in front of you, which limb would you use to kick the ball with?” The chosen limb was considered the dominant limb.

A Biodex Balance System (Figure 2) was utilized to indicate postural stability outcomes through a dynamic platform that deviated 20 degrees in multiple planes. It is a valid and reliable method for balance measurements (ICC = 0.83). Each participant was asked to maintain their balance while standing on the affected lower limb with the upper limbs beside the body. The test was performed three times, with a 10-s rest in between. The dynamic platform was unlocked between levels, where the difficulty level ranged from 6 to 1, with 1 being the most unstable level. The overall stability index (OSI), anteroposterior stability index (APSI), and mediolateral stability index (MLSI) were recorded as these parameters reflect postural stability [25,26].

The procedure started with supplying the necessary participant’s information into the Biodex software. The parameters of the test were then selected. The test was conducted with the following specifications: bare feet, standing on two legs, medium difficulty (level 5 with open eyes), 30-s trial time, 10-s rest interval, and one familiarization trial prior to each actual test. The subject was given instructions prior to the test to refrain from using their hands as a means of support and to maintain the platform as horizontal as possible by manipulating a cursor on the Biodex screen grid using visual feedback. After a 5-s delay, the platform was released after pressing the start key [27].

#### 2.4.2. Star Excursion Balance Test (SEBT)

SEBT measures dynamic balance as a shortage in dynamic postural control as a result of musculoskeletal injury (e.g., chronic ankle instability). The test initially encourages the maximal reaching of the opposite leg in eight directions while maintaining loading on a single leg; these directions relative to the stance leg were anterior (A), anteromedial (AM), medial (M), posteromedial (PM), posterior (P), postero-lateral (PL), lateral (L), and anterolateral (AL). Tape stuck on the floor and a protractor were used to mark the 8 directions with 45° in between each marking (Figure 3). 

The examiner asked the participants to stand on the affected ankle in the center of the tape and try to reach as far as possible with the contralateral leg in the needed direction. During the examination, the participants should maintain their balance and not touch the floor. The test was repeated three times, and the average was calculated. The test was found to be valid and reliable (ICC = 0.84–0.92) [28,29].

### 2.5. Sample Size

The sample size was calculated using the G*power 3.0.10 software (Heinrich Heine University Düsseldorf, Düsseldorf, Germany) depending on the Biodex stability index (OSI) score extracted from a previous study [20], and assuming 80% power and α = 0.05, a sample size of 81 participants was found to be adequate. In anticipation of a 10% dropout rate, the final sample size was increased to 90 (30 in each group).

### 2.6. Randomization 

Permuted blocks with variable sizes were used to randomly allocate the participants into the three arms of intervention. The order of randomization was performed online using the www.sealedenvelope.com website. 

### 2.7. Allocation

For concealment purposes, a code number was given to each subject, indicating the interpretation of the allocation sequence. The participants, the outcome assessor, and the therapist were not aware of the allocation sequence mechanism. The principal investigator, a researcher who was involved in neither the assessment nor the treatment procedures, performed this step.

### 2.8. Blinding 

The participants and outcome assessor were kept blind in this study. The participants were not aware of the outcome of interest in this study, and the assessors were not aware of the allocation mechanism. Moreover, at the follow-up assessment, the participants in the KTG and MG were instructed to remove the tape before undertaking the outcome assessment. 

### 2.9. Statistical Methods

All outcome measurements were computed using The Statistical Package for the Social Sciences (SPSS) Version 23.0; descriptive statistics were performed, while the means and standard deviations of the outcomes were checked using the Shapiro-Wilk test for normal distribution. A one-way repeated-measures ANOVA was used to assess the within- and between-group differences. In the event that a significant difference was determined between groups, subsequent post hoc tests were performed to determine the source of the difference. Cohen’s *d* formula was used to determine the clinical effect size, where values between 0.3 and 0.79 were considered medium, values above 0.79 were considered high, and values below 0.3 were considered low. *p* < 0.05 was considered significant.

## 3. Results

Thirty participants were randomly assigned to each of the three groups. For personal reasons, two participants could not continue the study: one from the KTG and the other from the EG. None of the participants experienced increased pain, skin irritation, or any musculoskeletal injury during the period of the study. The participants were similar at baseline regarding demographic characteristics (Table 1). The flowchart of the study procedure is shown in Figure 4. 

### 3.1. Postural Indices Results

As described in Table 2, ANOVA revealed statistically significant differences between groups regarding all postural indices. The MG demonstrated statistically significant improvement with a high effect size for the OSI values compared to KTG or EG (*p* = 0.01, Cohen’s *d* = 1.6, and *p* < 0.001, Cohen’s *d* = 1.63, respectively). Additionally, the KG demonstrated statistically significant improvement with a medium effect size compared to the EG (*p* = 0.04, Cohen’s *d* = 0.7).

The MG demonstrated a statistically significant improvement with a high effect size for the APSI values compared to the KG or EG (*p* = 0.02, Cohen’s *d* = 0.95, and *p* < 0.001, Cohen’s *d* = 1.22, respectively). The KTG and EG values of the APSI were not statistically significantly different (*p* = 0.21). 

Regarding the MLSI, the KTG demonstrated a statistically significant improvement with a medium to high effect size compared to MG or EG (*p* = 0.04, Cohen’s *d* = 0.6, and *p* < 0.01, Cohen’s *d* = 0.96, respectively). On the other hand, both the KTG and MG demonstrated no significant (*p* = 0.72) or clinically meaningful difference (Cohen’s *d* = 0.28).

The within-group analysis showed statistically significant improvements in all indices, with the highest mean differences in the OSI of the MG (MD = 0.18), the APSI of the MG (MD = 0.17), and the MLSI of the KTG (MD = 0.19).

### 3.2. Dynamic Stability Results

As shown in Table 3, the between-group analysis demonstrated statistically significant values and a high effect size for the SEBT in the posterior (*p* = 0.002, Cohen’s *d* = 1.2) and lateral (*p* < 0.02, Cohen’s *d* = 0.92) directions for the MG compared to KTG and EG.

The within-group analysis showed that the values of dynamic stability were statistically significantly increased in all groups and all directions except for the posterior direction in the KTG (*p* = 0.22 and MD = 3.93), and the lateral (*p* = 0.33 and MD = 4.27) and anterolateral (*p* = 0.19 and MD = 6.87) directions in the EG. 

## 4. Discussion

This study was conducted to compare the effect of KT versus exercises on postural stability and dynamic stability in RRs with ankle instability. The results demonstrated statistically significant improvements in postural stability indices (OSI, APSI, and MLSI) after treatment. Comparing the results between groups, the MG demonstrated the best results with both statistical significance and high effect sizes, while the KT group demonstrated better values compared to the EG. 

Regarding the SEBT results, most of the scores improved at the end of the intervention. However, variable results were obtained after comparing groups. The statistically significant differences with high effect sizes were evident in the dynamic stability in MG regarding the posterior and lateral directions. 

The improvement reported in the current study might be attributed to several reasons. KT stimulates the skin over which it is applied. KT creates a pull on the skin that continues for a long period of time, consequently giving consistent proprioceptive data to the territory of the body it covers [30]. This skin stimulation has been linked to the excitation of proprioception and improvement in the activities of the tibialis posterior muscles [30]. Consequently, tibialis posterior muscle activation time improves and reduces the susceptibility to inversion stress [31]. The stimulatory effect of Kt on muscle strength has been examined previously [32]. According to Dogan and colleagues [16], a physiologic examination showed that KT has positive effects on neural activation and can enhance muscle power, especially when applied for a certain period of time and augmented with therapeutic training.

Moreover, KT might enhance the judgment of position and orientation by providing cutaneous sensory stimulation [33] and consequently improve the ability to maintain stability and reduce postural sway so that the values of postural indices improve and the person can achieve higher values on SEBT. 

Although taping of the ankle has been studied in many aspects [15,16,20,34], RRs have received little interest from researchers. Stocco et al. [35] examined the effectiveness of KT applied with progressive tension on knee muscle strength in runners. The duration of the intervention in this study was similar to ours (eight weeks). The strength of the muscles was assessed using an isokinetic dynamometer. Interestingly, Stocco et al. did not observe any significant differences between participants who received KT and the control (placebo) group. This finding could be attributed to the small sample size used (n = 49). 

Siu et al. [36] applied KT to prevent foot pronation in runners with functional flat feet. Siu and colleagues asked the participants to practice running while they were taped and evaluated the behavior of their feet. The results of the Siu study concluded that KT could facilitate tibialis posterior function and support the transverse arch of the foot immediately after application. They also reported an increase in tibialis anterior muscle activity, especially during the first 15 min of running.

In a recent study [37], dynamic postural control was assessed using the modified SEBT in a group of patients with chronic ankle instability. KT was compared to those with bare feet and those who underwent dynamic taping. The results demonstrated significantly higher reach distances in the group that received KT compared to the other participants, especially in the medial and postero-medial directions. These findings support those reported in an earlier study [23]. 

Khalili et al. [38] compared the effect of adding KT to balance exercises on the balance (indicated by OSI) and stability scores (indicated by CAIT) of athletic female subjects. After 6 weeks of training, the OSI and stability scores were significantly better in the KT group compared to those of the control group. According to Khalili, the OSI is the main postural index that can accurately represent the overall ability of subjects with ankle instability to maintain their balance. In the current study, similar findings were reported; however, using KT with exercises was a more effective approach. 

Subjects who practice recreational running and are exposed to repeated ankle injuries should apply KT using the technique adopted in the current study and practice therapeutic exercises containing balance training. This way, the RRs improve sensory input from their injured ankles, enhance muscle power, and may decrease the risk of injury recurrence, so they can continue their sport for a longer time. Professional subjects and health authorities should conduct educational workshops and provide appropriate flyers to increase the RRs awareness regarding the role of KT and exercises to train subjects how to apply KT using proper guidelines.

In the current study, we could not blind the therapist to the interventions, which could affect the quality of administering the treatment to different groups. However, the extensive experience of the therapist could reduce this effect. The outcome measures were obtained before and after the end of the intervention period only. Additionally, these outcomes might not be sufficient to reflect the detailed nature of the ankle status of the study population. Hopefully, future studies will incorporate an intermediate- and long-term follow-up to investigate the lasting effect of the interventions used and implement more outcome measures.

## 5. Conclusions

Kinesio taping plus exercises improved postural stability compared with the exercises alone or kinesio taping alone. Kinesio taping and exercises improved dynamic stability, as measured using the star excursion balance test. Adding kinesio taping to exercises is a beneficial treatment intervention for recreational runners with ankle instability for the prevention and treatment of ankle sprains in people with chronic ankle instability. Recreational runners with ankle instability should be educated about practicing balance exercises and applying Kinesio taping.

## Figures and Tables

**Figure 1 healthcare-11-01749-f001:**
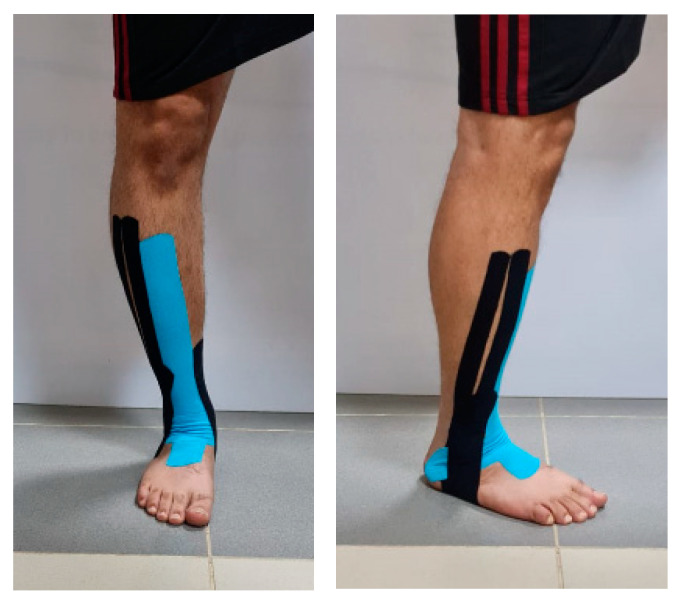
Taping technique of the ankle.

**Figure 2 healthcare-11-01749-f002:**
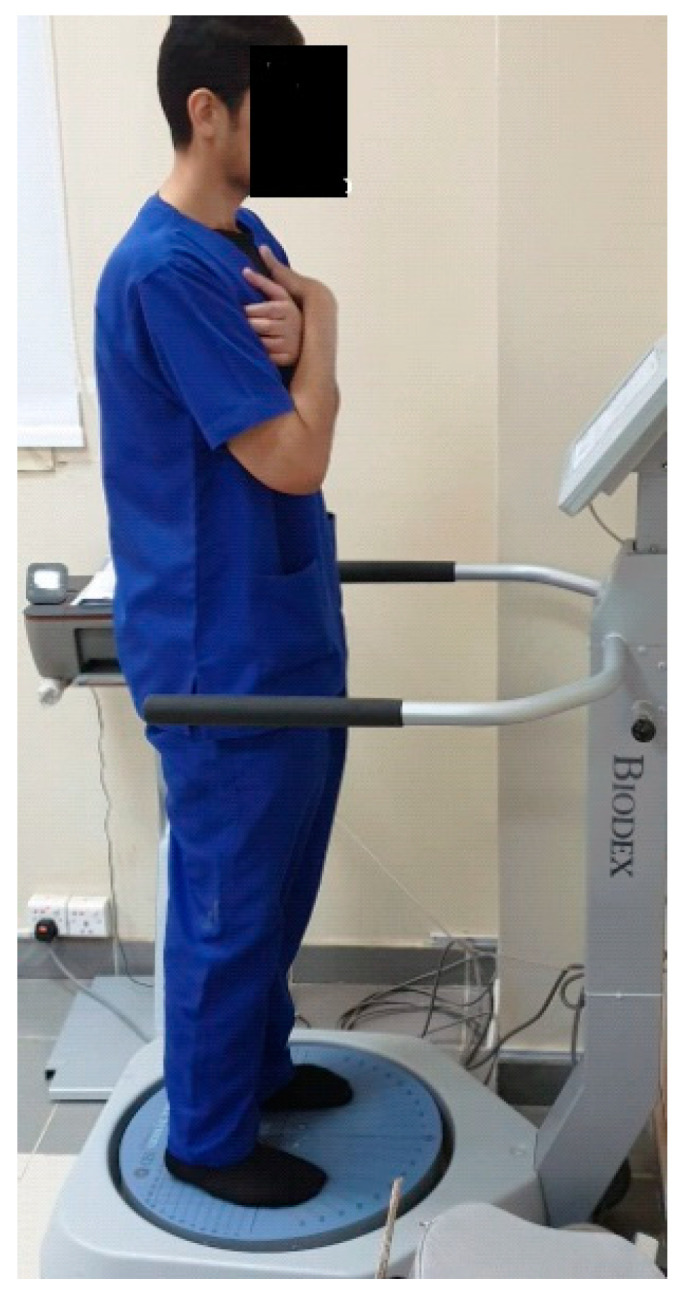
Assessing postural stability indices using the Biodex balance system.

**Figure 3 healthcare-11-01749-f003:**
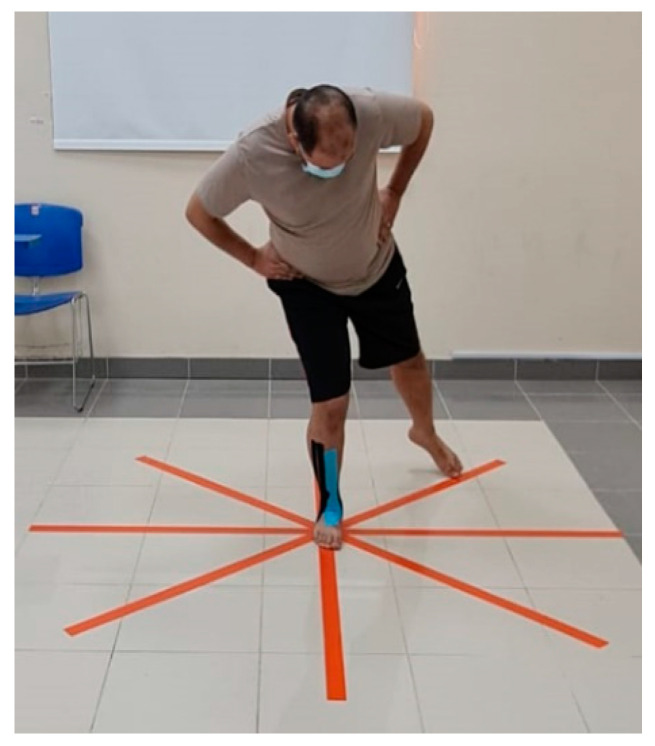
SEBT test.

**Figure 4 healthcare-11-01749-f004:**
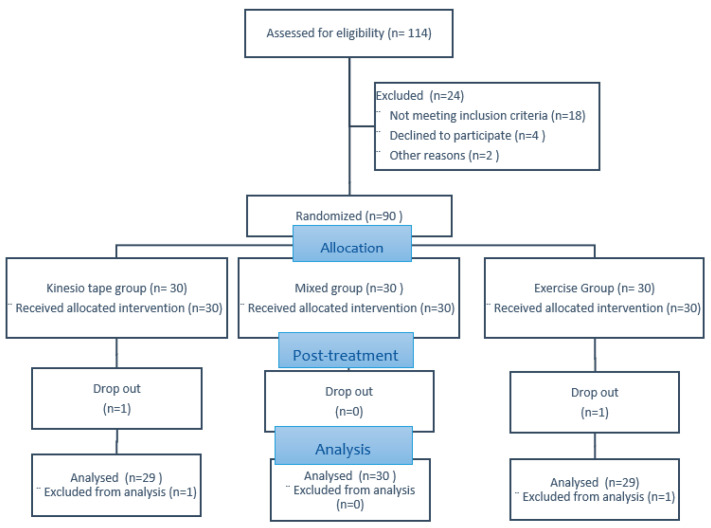
Flowchart of the study procedure.

**Table 1 healthcare-11-01749-t001:** Baseline characteristics of participants.

Variables	KTG n = 29	MG n = 30	EG n = 29	*p*
	Mean ± SD.	Mean ± SD.	Mean ± SD.	
Age (years)	32.69 ± 7.13	29.27 ± 8.65	31.87 ± 7.79	>0.05
Weight (kg)	62.0 ± 10.81	61.6 ± 5.96	62.6 ± 6.85	
Height (cm)	159.7 ± 3.47	159.53 ± 6.01	156 ± 5.91	
BMI	24.30 ± 4.20	24.28 ± 2.97	25.78 ± 3.16	
CAIT	25.8 ± 1.47	25.4 ± 1.63	24.9 ± 1.74	
Male/female n (%)	25(86.2)/4(13.8)	24(80)/6(20)	24(82.7)/5(17.3)	
Rt (%)/Lt (%) dominance	25(86.2)/4(13.8)	25(83.3)/5(16.6)	28(96.6)/1(3.4)	

KTG, kinesio taping group; MG, mixed group; EG, exercise group; SD, standard deviation; Kg, kilogram, cm, centimeter; BMI, body mass index, CAIT, Cumberland ankle instability tool; n, number; Rt, right dominance; Lt, lift dominance; *p*, probability value.

**Table 2 healthcare-11-01749-t002:** Within- and between-group comparisons for postural indices.

		KTG n = 29	MG n = 30	EG n = 29	ARMS (*p* and Cohen’s *d*)
		Mean ± SD.	Mean ± SD.	Mean ± SD.	
OSI	Pre	0.35 ± 0.14	0.42 ± 0.19	0.38 ± 0.22	KTG-MG: 0.113 (0.70) KTG-EG: 0.19 (0.45) MG-EG: 0.777 (0.17)
	Post	0.14 ± 0.05	0.23 ± 0.05	0.30 ± 0.12	KTG-MG: 0.015 * (1.60) KTG-EG: 0.000 * (1.63) MG-EG: 0.040 * (0.70)
	*p*-value	0.000 *	0.008 *	0.017 *	
	MD	0.16	0.18	0.08	
APSI	Pre	0.46 ± 0.05	0.44 ± 0.06	0.42 ± 0.06	KTG-MG: 0.734 (0.26) KTG-EG: 0.311 (0.45) MG-EG: 0.179 (0.35)
	Post	0.33 ± 0.06	0.27 ± 0.07	0.36 ± 0.08	KTG-MG: 0.028 * (0.95) KTG-EG: 0.213 (0.30) MG-EG: 0.001 * (1.22)
	*p*-value	0.000 *	0.000 *	0.001 *	
	MD	0.12	0.17	0.06	
MLSI	Pre	0.38 ± 0.05	0.40 ± 0.06	0.36 ± 0.07	KTG-MG: 0.311 (1.13) KTG-EG: 0.799 (0.36) MG-EG: 0.446 (0.61)
	Post	0.19 ± 0.07	0.25 ± 0.11	0.28 ± 0.12	KTG-MG: 0.042 * (0.60) KTG-EG: 0.018 * (0.96) MG-EG: 0.728 (0.28)
	*p*-value	0.000 *	0.000 *	0.005 *	
	MD	0.19	0.15	0.08	

KTG, kinesio taping group; MG, mixed group; EG, exercise group; SD, standard deviation; OSI, overall stability index; APSI, anteroposterior stability index; MLSI, mediolateral stability index; SEBT, star excursion balance test; MD, mean difference; *, significant as *p*-value < 0.05.

**Table 3 healthcare-11-01749-t003:** Within- and between-group comparisons for the star excursion balance test.

Test		KTG	MG	EG	ARM (*p* and Cohen’s *d*)
		Mean ± SD.	Mean ± SD.	Mean ± SD.	
Anterior	Pre	66.07 ± 12.69	68.73 ± 13.01	66.4 ± 15.23	KTG-MG: 0.597 (0.208) KTG-EG: 0.947 (0.024) MG-EG: 0.643 (0.165)
	Post	75 ± 11.88	76.73 ± 10.03	75.53 ± 9.86	KTG-MG: 0.658 (0.158) KTG-EG: 0.891 (0.049) MG-EG: 0.759 (0.121)
	*p*-value	0.000 *	0.001 *	0.001 *	
	MD	8.93	8.00	9.13	
Antero-medial	Pre	59.47 ± 11.24	62.47 ± 12.26	63.87 ± 14.6	KTG-MG: 0.524 (0.255) KTG-EG: 0.351 (0.338) MG-EG: 0.766 (0.104)
	Post	69.87 ± 10.41	70.73 ± 13.23	72.93 ± 14.37	KTG-MG: 0.854 (0.073) KTG-EG: 0.515 (0.244) MG-EG: 0.640 (0.159)
	*p*-value	0.005 *	0.000*	0.001 *	
	MD	10.4	8.27	9.07	
Medial	Pre	62.53 ± 15.54	70.6 ± 16	67 ± 16.0	KTG-MG: 0.171 (0.211) KTG-EG: 0.445 (0.283) MG-EG: 0.537 (0.225)
	Post	72.87 ± 13.9	79.47 ± 16.63	75.2 ± 16.79	KTG-MG: 0.260 (0.301) KTG-EG: 0.689 (0.151) MG-EG: 0.465 (0.255)
	*p*-value	0.000 *	0.000 *	0.001 *	
	MD	10.33	8.87	8.20	
Postero-medial	Pre	59 ± 11.58	65.2 ± 12.01	67.8 ± 13.19	KTG-MG: 0.174 (0.126) KTG-EG: 0.056 (0.249) MG-EG: 0.565 (0.206)
	Post	70 ± 9.82	74.53 ± 15.06	78 ± 16.97	KTG-MG: 0.389 (0.357) KTG-EG: 0.132 (0.577) MG-EG: 0.510 (0.216)
	*p*-value	0.000 *	0.001 *	0.007 *	
	MD	11	9.33	10.2	
Posterior	Pre	57.67 ± 9.39	58.8 ± 12.4	58.13 ± 11.04	KTG-MG: 0.908 (0.046) KTG-EG: 0.779 (0.103) MG-EG: 0.869 (0.057)
	Post	61.6 ± 12.59	78.67 ± 15.68	68.13 ± 14.74	KTG-MG: 0.002 * (1.2) KTG-EG: 0.221 (0.477) MG-EG: 0.052 (0.692)
	*p*-value	0.223	0.021 *	0.004 *	
	MD	3.93	20.53	9.33	
Postero-lateral	Pre	49.13 ± 11.71	55.07 ± 12.85	47.93 ± 12.34	KTG-MG: 0.194 (0.283) KTG-EG: 0.791 (0.131) MG-EG: 0.120 (0.366)
	Post	56 ± 12.49	62.13 ± 15.75	59.53 ± 19.3	KTG-MG: 0.303 (0.431) KTG-EG: 0.551 (0.217) MG-EG: 0.660 (0.148)
	*p*-value	0.000 *	0.001 *	0.004 *	
	MD	6.87	7.07	11.6	
Lateral	Pre	40.93 ± 10.65	52.0 ± 17.86	50.2 ± 16.35	KTG-MG: 0.054 (0.401) KTG-EG: 0.104 (0.222) MG-EG: 0.749 (0.105)
	Post	47.4 ± 10.53	60.13 ± 16.45	54.47 ± 16.03	KTG-MG: 0.021 * (0.922) KTG-EG: 0.192 (0.521) MG-EG: 0.294 (0.349)
	*p*-value	0.000 *	0.000 *	0.332	
	MD	6.47	8.13	4.27	
Antero-lateral	Pre	49.93 ± 10.27	52.8 ± 10.76	52.4 ± 12.74	KTG-MG: 0.491 (0.273) KTG-EG: 0.553 (0.213) MG-EG: 0.923 (0.034)
	Post	56.87 ± 10.33	58.47 ± 11.96	59.27 ± 11.18	KTG-MG: 0.697 (0.143) KTG-EG: 0.560 (0.223) MG-EG: 0.846 (0.069)
	*p*-value	0.000 *	0.001 *	0.19	
	MD	6.93	5.67	6.87	

KTG, kinesio taping group; MG, mixed group; EG, exercise group; SD, standard deviation; OSI, overall stability index; APSI, anteroposterior stability index; MLSI, mediolateral stability index; MD, mean difference; *, significant as *p*-value < 0.05.

## Data Availability

All data regarding this research work are fully available from the corresponding author upon request.

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
