# Peer review of "The Effect of Kinesio Taping on Balance and Dynamic Stability in College-Age Recreational Runners with Ankle Instability"

_healthcare, 2023, doi:10.3390/healthcare11121749_

Round 1

Reviewer 1 Report

Although it is an interesting work with a large sample, work is needed in the contextualization and in the reasoning of the discussion. In addition, the definition of the lesion is not clear. It does not provide a comprehensive vision of treatment, or at least this information should be perfectly defined in the selection criteria.

Abstract does not provide an overview of the most important aspects of the design and the results. the conclusions do not respond to the objectives. 

In the introduction, the background is focused on another lesion, nor is there any mention of the relationship between this lesion and the one addressed in the study. It does not even appear in the inclusion criteria. 

There is no consistency in the definition of whether it is an injury or a risk factor, since the summary and the introduction say different things. It is not understood that the introduction focuses on a different lesion. 

The introduction is too long and does not address the nature of the study. 

The references are generally more than 10 years old, with the exception of 3-4. I recommend using the latest available evidence on each aspect. 

Only 2 variables are assessed, which although important, do not determine the status of the lesion. This is especially true for a 12 week intervention. 

The premise of the use of the technique is applied to another lesion, which, on the other hand, does not appear in the inclusion criteria nor is its use in the population justified. 

What is the logical reasoning behind applying kinesio-taping for 8 weeks, and what possible disadvantages could it have?

It is not clear which group performs the proprioceptive exercises (it is only mentioned that 2 groups will do conventional exercises). If added to the conventional exercises, do they do them together or on different days?

In the Star Excursion Balance test, mobility has a strong influence (especially in anterior movements). This aspect is not mentioned at any point. It would be useful to talk about it and try to argue it in the discussion, applying a critical reasoning of the results, among other aspects that are lacking in the discussion. 

At one point in the paper it is mentioned that there is a follow up, but this does not appear again, so I do not know if this is an error, and the authors are refer to "post-treatment", or if there are really more results that the authors have not provided (at clinicaltrials.gov they mentioned that the time frame of the outcome mesures is 4 weeks, so it's a bit confusing)

There is a lack of information on the results in the tables, especially those related to the variables between groups. Percentages in demographic outcomes by sex are not shown 

The balance and SEBT variables isolated do not provide much clinical information in the complete approach to the lesion. They would specify deeply this point at the introduction, and after resolving it at the discussion.

Author Response

Reviewer 1

Although it is an interesting work with a large sample, work is needed in the contextualization and in the reasoning of the discussion. In addition, the definition of the lesion is not clear. It does not provide a comprehensive vision of treatment, or at least this information should be perfectly defined in the selection criteria.

Reply:

- The entire manuscript has been subjected to extensive rewriting and reconstruction according to reviewers’ comments.

- also, English editing was performed

- more information were added to the abstratct, introduction, and discussion sections to improve the contextualization

- the inclusion criteria were revised (lines 125-126).

- Abstract does not provide an overview of the most important aspects of the design and the results.

Reply:

- Authors added more information to the Methods section of the Abstract to clarify the design of the study, outcome measures, and the type of intervention. Please refer to lines  21 - 27

- the conclusions do not respond to the objectives. 

- conclusion was rewritten to focus on the scope of the study. Please refer to lines: 36-39

In the introduction, the background is focused on another lesion, nor is there any mention of the relationship between this lesion and the one addressed in the study.

It does not even appear in the inclusion criteria. 

Reply:

-Thank you for this comment. The authors added more information to highlight the link between ankle sprain and instability, and the consequences of these injuries on participants of recreational running. Please refer to line 43-99.

- the main lesion which is ankle instability is already mentioned in the inclusion criteria. Please refer to lines 125-126

There is no consistency in the definition of whether it is an injury or a risk factor, since the summary and the introduction say different things. It is not understood that the introduction focuses on a different lesion. 

Replay:

The authors apologize for this conflict, we tried to talk extensively about the lateral ankle sprain because it is the most common injury that might lead to ankle instability. But this led to neglection of talking about the main problem. Authors modified many parts in the introduction section in response to this comment. For example, a description of ankle instability was added, and the flow of data was improved so that the reader can follow the purpose of the research.  Please refer to the entire introduction section .

The introduction is too long and does not address the nature of the study. 

Reply:

- we totally agreeing that the introduction is long. However, the journal policy is to have 4000 words, so we had to. After revision, the nature of the study was highlighted in many sites throughout the introduction section.

The references are generally more than 10 years old, with the exception of 3-4. I recommend using the latest available evidence on each aspect. 

Reply:

Authors tried hard to update most of the old references. Please refer to the reference list

Only 2 variables are assessed, which although important, do not determine the status of the lesion. This is especially true for a 12 week intervention. 

Reply:

- Although the authors assessed only 2 main outcomes. But in fact, it can be considered 4 (3 postural indices and star excursion test). Additionally, the readings of the postural indices give valuable information about the sensory integration, proprioception, and postural reactions of the body which are all can be easily affected by ankle injuries. Many other researchers used similar kind of outcome measures, for example, Khalili et al 2022 (https://doi.org/10.3390/life12020178) measured one postural index (OSI) only the CAIT. Another example is the Lim et al 2021 (https://doi.org/10.12674/ptk.2021.28.3.200 ) where they depended on the SEBT test.

- regarding the duration of the intervention, it was 8 weeks not 12. This intervention period has been used in several studies as Khalili study (6 weeks)

The premise of the use of the technique is applied to another lesion, which, on the other hand, does not appear in the inclusion criteria nor is its use in the population justified. 

Reply:

the taping technique used in this study is already prescribed to protect lateral ankle sprains which is the common ankle injury leading to repeated injury and worsens the ankle stability level. All subjects participated in this study has been exposed to 1 or more lateral ankle sprains before the time of the study (we added this statement in the inclusion criteria please refer to lines 111-112).

What is the logical reasoning behind applying kinesio-taping for 8 weeks, and what possible disadvantages could it have?

Reply:

The authors intended to study the effect of using the KT for this long period to study the accumulative effect of using KT alone or during the performance of exercises as we did in the mixed group.

In the current study, subjects received KT underwent scheduled changing of the tape every 3 day so that the therapist can check for any adverse effects and also to allow for cleaning the area before retaping.

Using kinesiotape for a long period as we did is not uncommon. Others did the same as the following studies:

  • https://doi.org/10.1177/0269215520971764
  • DOI: 1097/PHM.0000000000000492
  • https://doi.org/10.53730/ijhs.v6nS6.10919

It is not clear which group performs the proprioceptive exercises (it is only mentioned that 2 groups will do conventional exercises). If added to the conventional exercises, do they do them together or on different days?

Reply:

Proprioceptive exercises are  part of the standard (conventional treatment) which was performed by MG and EG. This is clearly stated in lines 106 – 109.

MG were doing exercises while they taped. The tape was changed every 4 days and a new tape was applied till the end of the study. This was mentioned in lines 132-134

In the Star Excursion Balance test, mobility has a strong influence (especially in anterior movements). This aspect is not mentioned at any point. It would be useful to talk about it and try to argue it in the discussion, applying a critical reasoning of the results, among other aspects that are lacking in the discussion. 

Reply:

Discussion was subjected to extensive rewriting to address all missed information. Please refer to lines 318-383

At one point in the paper it is mentioned that there is a follow up, but this does not appear again, so I do not know if this is an error, and the authors are refer to "post-treatment", or if there are really more results that the authors have not provided (at clinicaltrials.gov they mentioned that the time frame of the outcome mesures is 4 weeks, so it's a bit confusing)

Reply:

- The follow up word was mentioned in the flow chart, and we used it to refer to the post treatment here. We changes the words to be post treatment to improve clarity (Figure 4).

- We apologize about this confusing, the intervention time posted in the clinical trial has been changed after the registration so its 8 weeks not 4 and there is no follow up data to report.

There is a lack of information on the results in the tables, especially those related to the variables between groups. Percentages in demographic outcomes by sex are not shown. 

Reply:

Thank you for this important comment. All missing data were added after revision of the SPSS results. Missed demographic data, post hoc analysis, effect size, and mean difference were added (please refer to tables 1 – 3)

The balance and SEBT variables isolated do not provide much clinical information in the complete approach to the lesion. They would specify deeply this point at the introduction, and after resolving it at the discussion.

Reply:

Authors agree with the reviewer’s comment. However, outcomes still provide valuable information regarding the performance of the ankle joint, sensori-motor function, ability to control the muscles around the ankle to maintain stability and achieve maximum reach. The reviewer’s concern was added to the limitation section in the discussion (lines 341-346).

Reviewer 2 Report

Thank you for the opportunity to review this manuscript.

The authors provide a study comparing kinesio-taping, kinesio-taping and exercise or exercise only effects on balance and maximum reach in recreational runners with ankle instability. The information is presented in an organized way but the more specific literature analysis and discussion would be helpful to readers. Also it would be beneficial to add information about clinical relevance in results. My suggestion is to calculate the effect size as indicator of clinical relevance in this study.

It would be useful to describe the concept of recreational runner in the introduction.

What do the authors mean by college-age? Is it specified in some reference?

Statistical methods. Were all the data distributed normally? For comparing three or more groups, consider using One-way ANOVA followed by an appropriate post hoc test, such as Tukey's HSD or other methods, rather than One-way repeated measures ANOVA with LSD.

Results. Please provide the information about participant numbers in each group in the tables.

Moderate editing of English language is recommended, since the grammar, spelling and style errors appears in the manuscript.

Author Response

Reviewer 2

Thank you for the opportunity to review this manuscript.

The authors provide a study comparing kinesio-taping, kinesio-taping and exercise or exercise only effects on balance and maximum reach in recreational runners with ankle instability. The information is presented in an organized way but the more specific literature analysis and discussion would be helpful to readers. Also it would be beneficial to add information about clinical relevance in results. My suggestion is to calculate the effect size as indicator of clinical relevance in this study.

 Reply:

- Thank you for your valuable comments. As you requested, the authors added more literature, especially to the introduction and discussion sections in order to enrich the discussion and to follow the comments of all reviewers.

- effect size was calculated, added to Table 2 and 3 and discussed in the Results section.

It would be useful to describe the concept of recreational runner in the introduction.

Reply:

Thank you for this comment, the general concept of recreational runners was added. Please refer to lines 15-18

What do the authors mean by college-age? Is it specified in some reference?

Reply:

The college-age is the usual age of the students who enter college which is usually between 17 and 24 years.  It was mentioned in many studies from various disciplines. The following are some examples:

  • https://doi.org/10.1080/07448481.2019.1590368
  • https://doi.org/10.1016/j.buildenv.2021.107909
  • https://doi.org/10.1016/j.nutres.2019.12.001

Statistical methods. Were all the data distributed normally? For comparing three or more groups, consider using One-way ANOVA followed by an appropriate post hoc test, such as Tukey's HSD or other methods, rather than One-way repeated measures ANOVA with LSD.

Reply:

Thank you for this important comment. We had to revise our statistical tests again. We found that all data were normally distributed.

We conducted  the post hoc tests and put the final results in tables 2 and 3. Also the mean differences and effect size (Cohen’s d value) were added to the table (2 and 3)

Results. Please provide the information about participant numbers in each group in the tables.

Reply:

Done. Please refer to the tables

Comments on the Quality of English Language

Moderate editing of English language is recommended, since the grammar, spelling and style errors appears in the manuscript.

Reply:

The entire manuscript was subjected to professional English editing using services provided by the journal.

Reviewer 3 Report

Comments to the Author

The authors of this article did an admirable job on an important topic, aimed to investigate the effect of kinesio-taping (KT) on balance and maximum reach in recreational runners with ankle instability. However, there are several points that require further clarity;

1- Page 1, Lines 22-23: The results section in the abstract requires revision to enhance clarity and incorporate numerical data. It would be beneficial to include specific information such as the pre-post percentage changes (differences) of OSI, APSI, or other parameters investigated in the study, as well as comparable percentage changes between groups.

2- Page 1, Lines 29-36: Please summarize this paragraph in one or two sentences and combine it with the next paragraph.

3- Page 2, Lines 57-65: I suggest you revise these two paragraphs to make them better and more fluent.

4- Page 2, Lines 66-75: It is worth noting that several studies have explored the effects of KT on various muscle groups and examined its impact on variables such as balance, strength, and performance. Here are a few studies for your paper. If you like, you can cite it:

https://doi.org/10.3389/fphys.2023.1064625

https://doi.org/10.23751/pn.v23iS1.11489

https://doi.org/10.1055/a-2035-8005

5- Page 2, Lines 76-79: Remove this paragraph

6- Page 2, Line 88: In the study, it is important to consider the participants' sports background as well as the proportions of dominant and non-dominant legs within the population. Additionally, it would be helpful to clarify on which feet the exercises were performed. Furthermore, it would be relevant to provide details on whether ankle instability was present in the same leg for all participants.

7- Page 3, Line 122: How was this method created? Can you reference it?

8- Page 4, Lines 150, 189, 195, 199, 205: Please move this section to the study design section and provide more systematic information about when, where, and how many visits the overall measurements of the study will take place. Additionally, I recommend combining the sections on Sample Size, Randomization, Allocation, and Blinding, placing them appropriately within the study design section to ensure clarity and coherence.3- Pages 1-2, Lines 26-93: The introduction is partially sufficient, but the physiological (cardiovascular disease, decreased pulmonary capacity, etc) and psychological (depression, anxiety, self-confidence or stress, etc) adverse effects of obesity need to be mentioned more.

9- Page 6, Line 211: The method should be explained in more detail. There are also some deficiencies. For example, the method used for pre-post tests is not written.

10- Page 7, Lines 233-240: I suggest you shorten the sentences and point out the main differences. Long sentences are difficult for readers and can have a negative impact on comprehensibility.

11- Pages 7-8, Lines 242, 252: It might be better to give the pre-post test percent improvements of each group. Please indicate percentage differences in the table.

12- Page 7, Lines 246-251: I suggest you shorten the sentences and point out the main differences.

13- Page 9, Line 256: The discussion needs to be seriously revised and supported by more comprehensive literature. This section is often the most difficult part to write. However, following some basic rules will help. The Discussion should be formatted like this:

First paragraph: Summarizing the aim of the study and the main results

Second paragraph: Discussing potential mechanisms and explanations for the findings

Third paragraph: Summarizing previous studies in the field and compare them with the present findings

Fourth: Listing strengths and potential limitations

Fifth (if applicable): Discussing clinical implications of the findings

Sixth: The Conclusion

14- Page 10, Line 316: I think this section deserves more than one sentence.

GENERAL COMMENTS:

1. The manuscript requires language improvement.

2. The topic is important but especially the introduction and discussion sections should be improved significantly. Literature review is nonadequacy.

3. Abstract should be re-edited after changes made in the article.

The language needs improvement.!

Author Response

Reviewer 3

Comments and Suggestions for Authors

Comments to the Author

The authors of this article did an admirable job on an important topic, aimed to investigate the effect of kinesio-taping (KT) on balance and maximum reach in recreational runners with ankle instability. However, there are several points that require further clarity;

1- Page 1, Lines 22-23: The results section in the abstract requires revision to enhance clarity and incorporate numerical data. It would be beneficial to include specific information such as the pre-post percentage changes (differences) of OSI, APSI, or other parameters investigated in the study, as well as comparable percentage changes between groups.

Reply:

Done please refer to lines 27-36

2- Page 1, Lines 29-36: Please summarize this paragraph in one or two sentences and combine it with the next paragraph.

Reply:

- according to the Editors we still need to add 500 words to meet the minimum number of words (4000) so we tried to refine the style of writing to keep the number of words and improve fluency  us to increase the length of the study as the manuscript is 500 words shorter than recommended. We tried to reorganize the writing to keep the length and add the missed data regarding the nature of the study.

- also we followed the instruction and advices of the professional English editors

3- Page 2, Lines 57-65: I suggest you revise these two paragraphs to make them better and more fluent.

Reply:

Thanks for this comment: the whole manuscript has been subjected to professional English editing and many parts exposed to rewriting and paraphrasing.

4- Page 2, Lines 66-75: It is worth noting that several studies have explored the effects of KT on various muscle groups and examined its impact on variables such as balance, strength, and performance. Here are a few studies for your paper. If you like, you can cite it:

https://doi.org/10.3389/fphys.2023.1064625

https://doi.org/10.23751/pn.v23iS1.11489

https://doi.org/10.1055/a-2035-8005

Reply:

thank you very much for these suggestions. Authors cited 2 (Lin et al 2023 and Dogan et al 2021) of the three suggested papers.  

5- Page 2, Lines 76-79: Remove this paragraph

Reply:

Paragraph removed.

6- Page 2, Line 88: In the study, it is important to consider the participants' sports background as well as the proportions of dominant and non-dominant legs within the population. Additionally, it would be helpful to clarify on which feet the exercises were performed. Furthermore, it would be relevant to provide details on whether ankle instability was present in the same leg for all participants.

Reply:

- We already did. We excluded any subjects who regularly participate in any other sport. Related statement was added to the exclusion criteria, please refer to lines 125-126

- we measured the dominance, but we did not report data regarding the effect of dominance in the manuscript because we thought that it’s not an important factor for a subject who practices running as both lower limbs participate equally. BUT up to your request, data regarding dominance were added to table 1.

7- Page 3, Line 122: How was this method created? Can you reference it?

Reply:

This intervention method was conducted before by Clark & Burden 2005.  This reference is already cited in the text (please refer to line 154).

8- Page 4, Lines 150, 189, 195, 199, 205: Please move this section to the study design section and provide more systematic information about when, where, and how many visits the overall measurements of the study will take place. Additionally, I recommend combining the sections on Sample Size, Randomization, Allocation, and Blinding, placing them appropriately within the study design section to ensure clarity and coherence.

Reply:

- Regarding the general information of the outcomes presented in line 150, they were moved to the study design.

- regarding the other information reported in this comment, the authors believe that they are in the correct place according to the guidelines of the CONSORT statement for reporting randomized clinical trials.

3- Pages 1-2, Lines 26-93: The introduction is partially sufficient, but the physiological (cardiovascular disease, decreased pulmonary capacity, etc) and psychological (depression, anxiety, self-confidence or stress, etc) adverse effects of obesity need to be mentioned more.

Reply:

Due to another reviewer comment regarding the length of the introduction, authors cannot add more text especially that obesity was not considered in this study (participants were of normal or overweight category only)

9- Page 6, Line 211: The method should be explained in more detail. There are also some deficiencies. For example, the method used for pre-post tests is not written.

Reply:

more data was added to the statistical design section. Please refer to lines 246-251

10- Page 7, Lines 233-240: I suggest you shorten the sentences and point out the main differences. Long sentences are difficult for readers and can have a negative impact on comprehensibility.

Reply:

The entire manuscript subjected to professional English editing.

11- Pages 7-8, Lines 242, 252: It might be better to give the pre-post test percent improvements of each group. Please indicate percentage differences in the table.

Reply:

We added much information in tables such as mean difference of improvement, effect size, and post hoc test results

12- Page 7, Lines 246-251: I suggest you shorten the sentences and point out the main differences.

Reply:

The entire manuscript subjected to professional English editing.

13- Page 9, Line 256: The discussion needs to be seriously revised and supported by more comprehensive literature. This section is often the most difficult part to write. However, following some basic rules will help. The Discussion should be formatted like this:

First paragraph: Summarizing the aim of the study and the main results

Second paragraph: Discussing potential mechanisms and explanations for the findings

Third paragraph: Summarizing previous studies in the field and compare them with the present findings

Fourth: Listing strengths and potential limitations

Fifth (if applicable): Discussing clinical implications of the findings

Sixth: The Conclusion

Reply:

Thank you for your valuable guide. The discussion was reconstructed according to the comment and more recent literature with detailed information were added.

GENERAL COMMENTS:

  1. The manuscript requires language improvement.

Reply:

-professional English editing was performed

  1. The topic is important but especially the introduction and discussion sections should be improved significantly. Literature review is nonadequacy.

Reply:

Literature was updated and more information were added

  1. Abstract should be re-edited after changes made in the article.

Reply:

done

Comments on the Quality of English Language. The language needs improvement.!

 Reply:

The entire manuscript was sent for professional English editing before resubmission.

Round 2

Reviewer 1 Report

The authors have made corrections and improved the manuscript considerably. 

Author Response

thank you for your positive comment. we appreciate the effort and valuable contribution 

Reviewer 2 Report

Thank you again for the possibility to review this manuscript. It is clear that the manuscript has been sufficiently improved after comments of reviewers but some issues still remain and require additional improvements.

- Abstract: in the results part there is inappropriate use of abbreviations which makes it complicated to understand the abstract without knowing the full study design. Try to avoid using unexplained abbreviations like OSI, APSI, MLSI, SEBT or find a way how to use it without confusing a reader.

- I would suggest to change "maximum reach" to "dynamic stability".

- Materials and Methods. Participants: the right link for clinical studies database is https://clinicaltrials.gov, please correct it.

- Page 4, line 164: "Each sit was repeated <...>" I suppose it should be each SET.

- Results: the titles of tables 1,2 and 3 should be more specific, e.g. "Table 1. Characteristics of the participants at baseline." Please revise.

- Table 1. It is not clear what "Rt(%) / Lt(%)" means, please explain below the table.

- Results: the format of explanations of abbreviations below the tables are not the same in all the section. Table 2: "(KTG) kinesiotaping group, (MG) mixed group,(EG) exercise group, (SD) standard deviation, <...>". Please revise.

- Figure 4. Not sure what's the point of using this "Did not receive allocated intervention (=0)" if none of the participants were under this circumstances. Also there is a spelling mistake "Kinesiotap group", please revise and correct.

- Conclusions. In my opinion, it would be better not to use abbreviations in the conclusions of the whole study since.

Author Response

Reply to Reviewer 2

Thank you again for the possibility to review this manuscript. It is clear that the manuscript has been sufficiently improved after comments of reviewers but some issues still remain and require additional improvements.

- Abstract: in the results part there is inappropriate use of abbreviations which makes it complicated to understand the abstract without knowing the full study design. Try to avoid using unexplained abbreviations like OSI, APSI, MLSI, SEBT or find a way how to use it without confusing a reader.

Reply:

All abbreviations mentioned in the abstract were replaced by the full word expression. Highlighted in green.

- I would suggest to change "maximum reach" to "dynamic stability".

Reply:

Done. Corrections highlighted in green color.

- Materials and Methods. Participants: the right link for clinical studies database is https://clinicaltrials.gov, please correct it.

Reply:

Done, please refer to line 119

- Page 4, line 164: "Each sit was repeated <...>" I suppose it should be each SET.

Reply:

Done, please refer to line 166

- Results: the titles of tables 1,2 and 3 should be more specific, e.g. "Table 1. Characteristics of the participants at baseline." Please revise.

Reply:

The titles of the 3 tables were revised and paraphrased

- Table 1. It is not clear what "Rt(%) / Lt(%)" means, please explain below the table.

Reply:

The meaning of these abbreviations was added to the footnotes

- Results: the format of explanations of abbreviations below the tables are not the same in all the section. Table 2: "(KTG) kinesiotaping group, (MG) mixed group,(EG) exercise group, (SD) standard deviation, <...>". Please revise.

Reply:

Revised, please refer to the footnotes of the tables

- Figure 4. Not sure what's the point of using this "Did not receive allocated intervention (=0)" if none of the participants were under this circumstances. Also there is a spelling mistake "Kinesiotap group", please revise and correct.

Reply:

The flow chart corrected.

- Conclusions. In my opinion, it would be better not to use abbreviations in the conclusions of the whole study since.

Reply:

Authors did as requested. Please refer to the conclusion sections in the abstract and main text